# Helicobacter pylori resistance to antibiotics before and after treatment: Incidence of eradication failure

Oddmund Nestegard[1,2]*, Behrouz Moayeri[3], Fred-Arne Halvorsen[4], Tor Tønnesen[4], Sveinung Wergeland Sørbye[5], Eyvind Paulssen[2], Kay-Martin Johnsen[2], Rasmus Goll[2], Jon Ragnar Florholmen[2], Kjetil K. Melby[3,6]

1 Ringerike Hospital, Vestre Viken Health trust, Hønefoss, Norway, 2 Research Group Gastroenterology and Nutrition, Department of Clinical Medicine, UiT the Artic University, Tromsø, Norway, 3 Department of Microbiology, Oslo University Hospital, Oslo, Norway, 4 Drammen Hospital, Vestre Viken Hospital Trust, Drammen, Norway, 5 Department of Pathology, University Hospital of North Norway, Tromsø, Norway, 6 Institute for Clinical Medicine, Faculty of Medicine, University of Oslo, Oslo, Norway

* oddmuan@online.no

**Data Availability Statement:** All relevant data are within the manuscript and its Supporting Information files. Data are also available from the AiT-Artic University Norway (contact prof. Jon

## Abstract

### Background

Increasing prevalence of antibiotic resistance especially to clarithromycin and metronidazole has been observed in *Helicobacter pylori (H. pylori)*.

### Aim

To characterize the antimicrobial resistance pattern of *H. pylori* before and after treatment in a cohort of patients accumulated over a period of 15 years after an unsuccessful eradication treatment had been given comparing sensitivity data from patients with newly diagnosed *H. pylori* infection. A specific objective was to look for resistance to levofloxacin.

### Material and methods

Total of 50 patients newly diagnosed for *H. pylori* infection treated with omeprazole and amoxicillin/clarithromycin and 42 *H pylori* treatment-resistant patients treated with omeprazole and amoxicillin/levofloxacin were enrolled in this study. Cultures including antibiotic sensitivity testing were conducted according to standard laboratory routines and thus also in keeping with a European study protocol using E-test gradient strips or disc diffusion methods.

### Results

Clarithromycin resistance was more frequently observed in the *H. pylori* resistant group than in newly diagnosed *H. pylori* group (39% versus 11%). Regarding metronidazole the distribution was 70% versus 38%, and 8% versus 12% were resistant to tetracycline. No resistance was observed for amoxicillin. After re-treatment of patients belonging to the *H. pylori* treatment-resistant group, just two patient strains were recovered of which one harbored

Florholmen, jon.florholmen@unn, for researchers who meet the criteria for access to confidential data.

**Funding:** No. The study is funded by Research Department, Vestre Viken Hopital, Drammen The funders had no role in study design, data collection and analysis, decision to publish, or preparation of the manuscript.

**Competing interests:** There are not any commercial affiliations or competing interests in our study. This does not alter our adherence to PLOS ONE policies on sharing data and materials.

metronidazole resistance. In the group of newly diagnosed *H. pylori*, seven patients were culture positive by control after treatment. Two and three patient strains showing resistance to clarithromycin and metronidazole, respectively. None of the strains in our material was classified as resistant to amoxicillin and levofloxacin. Whereas 12% was resistant to tetracycline in the newly diagnosed before treatment.

## Conclusion

Clarithromycin resistance was more frequent in the *H. pylori* treatment-resistant group than *strains* from patients with newly diagnosed *H. pylori* infection. No resistance was observed to amoxicillin and levofloxacin. In such cases Therefore levofloxacin may be used provided in vitro sensitivity testing confirms applicability.

## Trial registration

ClinicalTrials.gov identifier: NCT05019586.

## Introduction

*Helicobacter pylori* (*H. pylori*) was discovered in 1983 [1] and has been identified as the main pathogenic factor for gastric and duodenal peptic ulcers [2]. Gastric cancer is the third most common cause of cancer-related death in the world. It is now wellestablished that *H. pylori*-infection predispose individuals toward gastric adenocarcinoma later in life. Gastric cancer is the third most common cause of cancer-related death in the world [3]. *H. pylori* has been classified as a class I carcinogen by the World Health Organization (WHO).

WHO classified in 2017 *H. pylori* among 12 families of bacteria that are the most resistant to antibiotics.

The increase in *H. pylori* resistance to antibiotics can revive the problem of gastric cancer. Treatment of the *H. pylori* infection cures patients with such ulcers, thus this disease is no longer a chronically recurring and disabling condition in the Western world. However the infection is still a great problem worldwide, especially in developing countries, where up to 50% of the population may be infected [4, 5].

Initially, the most effective eradication regimens had an efficacy of >90% [6]. Unfortunately after 20 years of infection treatment increasing antimicrobial resistance of *H. pylori* has been installed [7]. This includes resistance to clarithromycin and metronidazole due to increased use of these antibiotics for other conditions.

Clarithromycin resistance has especially had a major negative impact as it has been a main constituent of the recommended first-line triple therapy for *H. pylori* eradication. Metronidazole resistance is highly prevalent but can to some extent be overcome [8]. Thus, the increasing prevalence of antibiotic resistance has to be considered when designing rational therapeutic regimens for *H. pylori* infection [9, 10]. In areas with a known high degree of resistance to both clarithromycin and metronidazole, bismuth quadruple therapy is recommended [11].

Parallel to the increasing antibiotic resistance, a cohort of patients infected with treatment-resistant *H. pylori* has accumulated. These patients will be at risk for the recurrence of peptic ulcer disease. Few reports exist on the clinical or the microbial characterizations of this cohort of patients. However, some data are emerging in the search for a new effective antibiotic therapy, including levofloxacin [12]. Thus, the goal of this study was to characterize the

antimicrobial resistance pattern of *H. pylori* in a cohort of patients accumulated over a period of 15 years after an unsuccessful eradication attempt.

## Materials and methods

### Ethics statement

The study is approved by the Regional Commitee for Medical and Health Research Ethics
  (REC North ID: 2009/175-15) including storage of biological material.
  Consents from the participants are obtained in oral and written form

### Study population

**Enrollment and patient flow.** The data presented are from the main Chronical Infection of Helicobacter Pylori (CHRIHEP) study published elsewhere [13]. Patients were recruited from three different cohorts admitted to three different gastrointestinal units at Norwegian hospitals: i.e. The University Hospital of North Norway, Tromsø; Drammen Hospital, Drammen and Ringerike Hospital, Hønefoss, the latter two both being parts of Vestre Viken Hospital Trust. We report the data obtained from the two groups of interest with active *H. pylori* infection:

Group 1 *(H. pylori* treatment- resistant n = 42), were recruited from medical files from 1990 to 2012, and from patients that had been treated unsuccessfully two or more times for *H. pylori* infection with different regimens [13]. All persons completed treatment and controls

Group 2 (*H. pylori* untreated, n = 50) were outpatients referred to the Gastrointestinal unit for upper endoscopy due to gastrointestinal complaints and diagnosed as being *H. pylori*-infected for the first time [13]. There were 52 patients, two of them denied participation.

The diagnosis of *H. pylori* in biopsies was based on a positive result for either the *H. pylori* rapid urease test or detection of the presence of *H. pylori* in histological specimens, or both [13].

**Determination of antimicrobial sensitivity.** Bacterial strains, isolated between 2010 and 2016, were examined using a standardized technique according to a European study protocol [10]. All isolates were recovered from biopsies from the stomach lining in patients with suspected *H. pylori*-related disease. Identification was based on colony morphology, Gram staining and positive catalase, oxidase and urease tests. The strains were kept at -70°C. The minimum inhibitory concentrations (MICs) of metronidazole, amoxicillin, clarithromycin, tetracycline and levofloxacin were determined by E-test (bioMerieux, France). Levofloxacin representing the quinolone group of antibiotics. The inoculum was adjusted to McFarland standard 3. Mueller Hinton agar (Beckton Dickinson) with 10% horse blood was used. Agar plates were incubated for 72–96 hours in microaerobic conditions (Campygen, OXOID, UK). MIC values were read according to the instructions from the manufacturer.

**Treatment.** Patients in Group 1 (*H. pylori* treatment-resistant) have retreated with oral omeprazole 20 mg b.i.d., amoxicillin 1 g b.i.d, and levofloxacin 500 mg b.i.d. for 10 days. Six of the 42 patients were still *H. pylori* positive after treatment. In Group 2 (*H. pylori* naïve), ten of 50 patients (20%) were *H. pylori* positive after treatment of oral omeprazole 20 mg b.i.d, amoxicillin 1 g b.i.d., and clarithromycin 500 mg b.i.d. for 7 days [13]. Subjects in both groups underwent gastroscopy 3–6 months after treatment and biopsies were collected for the various examinations described above.

**Statistics.** Data are presented as counts (percentage). Calculations and Chi-squared tests were performed in IBM SPSS Statistics 24 (IBM Corporation, Armonk, New York, USA)

**Table 1. Examinations for *H. pylori* by conventional culture of gastric biopsies.**

| Study group | Group 1 | | Group 2 | |
|---|---|---|---|---|
| | Before treatment | After treatment | Before treatment | After treatment |
| Cultures | 42 | 37 | 50 | 42 |
| negative | 16 (38.1%) | 35 (94.5%) | 8 (16.0%) | 33 (78.6%) |
| positive | 26 (61.9%) | 2 (5,4%) | 42 (84.0%) | 9 (21.4%) |

Results from *H. pylori* cultures of gastric mucosal biopsies before or after treatment in the two study groups; both groups show highly significant treatment response (p<0.001 by Chi-square). Group 1: resistant *H. pylori*, Group 2: naïve *H. pylori*.

Numbers are n or n (%). For further details, see text.

## Results

A total of 92 patients, divided into 42 in Group 1 and 50 in Group 2, were enrolled according to the inclusion criteria cited above. Exclusion criteria was double platelet inhibition and non-compliance [13]. Results from bacterial cultures of gastric mucosal biopsies are shown in Table 1. Among patients with positive *H. pylori* tests, positive cultures were obtained from 26 of 42 and 42 of 50 patients in Group 1 and Group 2, respectively, before treatment in this study (visit 1). After treatment (visit 2), two of 37 and nine of 42 cultures were positive for *H. pylori*, respectively (Table 1). Two of the nine positive cultures in Group 2 did not yield valid results. In Tables 2 and 3 the detailed antimicrobial resistance data are shown both before (Visit 1) and after treatment (Visit 2).

At visit 1, in Group 1 clarithromycin resistance was more frequently observed than in Group 2 (38,5% versus 10,5%), for metronidazole the distribution was 69,5% versus 38,1%, for tetracycline the distribution was 8% versus 11,9%. No resistance was observed for amoxicillin. At visit 2, in Group 1 only two isolates were positive, where one isolate showed metronidazole resistance. At visit 2, in Group 2, seven cultures were positive, two and three with resistance for clarithromycin and metronidazole, respectively.

None of the strains in our material was classified as resistant to amoxicillin or levofloxacin.

## Discussion

In the cohort of patients with treatment failures (secondary resistance) to *H. pylori*, accumulated over 15 years, the most prevalent resistance pattern was observed against metronidazole (69,5%) and a somewhat lower prevalence against clarithromycin (38,5%), even less against tetracycline (8%) and none towards amoxicillin. These data are in keeping with data presented from other research groups and general surveillance of antibiotic sensitivity of *H. pylori* [14]. The prevalence of antibiotic resistance in newly diagnosed patients *H. pylori* infection (primary resistance) was 38,1% for metronidazole and 10,5% for clarithromycin i.e.half the numbers compared to what were observed in H.pylori infected patients having treatment failures.

**Table 2. Antimicrobial resistance in *H. pylori* cultures from.**

| Group 1 | Before treatment (n = 26) | | | After treatment (n = 2) | | |
|---|---|---|---|---|---|---|
| | Sensitive | Resistant | Intermediate | Sensitive | Resistant | Intermediate |
| Amoxicillin | 26 (100%) | 0 | 0 | 2 (100%) | 0 | 0 |
| Clarithromycin | 16 (61.5%) | 10 (38.5%) | 0 | 2 (100%) | 0 | 0 |
| Metronidazole | 7 (30.4%) | 16 (69.5%) | 0 | 1 (50%) | 1 (50%) | 0 |
| Tetracycline | 23 (92.0%) | 2 (8.0%) | 0 | 2 (100%) | 0 | 0 |

**Table 3.  Antimicrobial resistance in *H. pylori* cultures from.**

| Group 2 | Before treatment (n = 42) | | | After treatment (n = 9; valid = 7) | | |
|---|---|---|---|---|---|---|
| | Sensitive | Resistant | Intermediate | Sensitive | Resistant | Intermediate |
| Amoxicillin | 42 (100%) | 0 | 0 | 7 (100%) | 0 | 0 |
| Clarithromycin | 38 (89.5%) | 4 (10.5%) | 0 | 5 (71.4%) | 1 (14.3%) | 1 (14.3%) |
| Metronidazole | 25 (59.5%) | 16 (38.1%) | 1 (2.4%) | 4 (57.2%) | 3 (42.8%) | 0 |
| Tetracycline | 36 (85.7%) | 5 (11.9%) | 1 (2.4%) | 7 (100%) | 0 | 0 |

Tests for antimicrobial resistance in cultures of *H. pylori* in the to study groups, before and after treatment. Group 1: resistant *H. pylori*, Group 2: naïve *H. pylori*. Two of the cultures in Group 2 "After treatment" did not yield valid test results. Numbers are n or n (%).

Resistance for tetracycline was 11,9%. None of the strains in our study was classified as resistant to neither amoxicillin nor levofloxacin. Even with these antibiotic resistance patterns, we achieve eradication rates of 79% and 92% in the Hp naïve and Hp resistant groups, respectively. Thus, our study indicates that secondary *H. pylori* resistance to metronidazole and clarithromycin may be overcome by the application of new antibiotic combinations. Over the last years, reports from different parts of the world have been published dealing with antimicrobial resistance rates in *H. pylori* [15]. At present, WHO considers *H. pylori* to be an important threat to human health based on the global wide appearance and declining antibiotic sensitivity [16]. Comparing results from different reports may, however, be difficult as in many cases different methodologies have been applied [15]. In addition, some presentations do not specify whether the reported numbers reflect primary resistance alone or a mix of strains from both categories [15].

Furthermore, higher levels of *H. pylori* primary antibiotic resistance might be expected in general in countries where antibiotic usage is higher than for instance in the Nordic countries (ECDC report 2015) [17]. An European multicenter study reported in 1997 an average primary resistance rate in Europe of 9.9% to clarithromycin [18]. Another study from 2013 showed that resistance rates to many drugs currently in use correlated with local antibiotic consumption [19]. In this study, the primary resistance to clarithromycin varied between 5.6% and 36.6%. An analysis in 2019 on the antibiotic sensitivity pattern in *H. pylori* in South-Eastern Norway showed a clarithromycin resistance of < 10% [20]. This figure corresponds with data obtained on clarithromycin use in the same period [21]. These findings support the lesser occurrence of antibiotic resistance in *H. pylori* provided a corresponding restricted prescription of clarithromycin. A follow-up study on the relationship between antibiotic consumption and *H. pylori* resistance support this notion [14].

There is no indication of increasing primary resistance to clarithromycin in our catchment area over the last 10 years as the present study showed a MIC50 value (the MIC value that inhibits 50% of the isolates) of 0.023 for clarithromycin, compared to 0.047 (In-house data, N = 23) obtained approximately 10 years ago. Data on primary resistance in Europe was shown to be 24% for clarithromycin, 34% for metronidazole and 20% for levofloxacin [22].

The reported resistance to metronidazole varies globally. In Europe, the prevalence of metronidazole resistance in *H. pylori* is generally between 20 and 40% [11], whereas the prevalence in developing countries is known to be higher (50–80%) [23]. Differences may reflect the widespread use of metronidazole in these areas, methodological differences, or weak reproducibility of the analysis. In our study, 69,5% (Group 1) and 38,1% (Group 2) of the isolates were resistant to metronidazole when the adopted European methodology was applied, i.e., microaerobic conditions for the entire incubation period. However, as metronidazole requires anaerobic conditions to be activated, the Norwegian reference group for antibiotics has

recommended that the application of anaerobic conditions be limited to the initial 24 hours of the incubation period when performing susceptibility testing for metronidazole. Applying this approach, the resistance rate towards metronidazole fell from 22.5% to 7.8% in a recent study [10], indicating that a substantial number of the resistant strains might be clinically susceptible to metronidazole. The previously mentioned study from 2013 showed that primary resistance rate to metronidazole [18] varied between 28.6% and 43.8%. Thus, in our setting the resistance to antibiotics used for the treatment of *H. pylori* is low with the exception of metronidazole and partly clarithromycin. Moreover, *H. pylori* resistance to clarithromycin has been reported from other countries leaving the use of levofloxacin for primary treatment resistant cases [24]. Levofloxacin resistance is due to changes in gyrA component of *H. pylori* [22–25] that renders even fluoroquinolones inactive to suppress *H. pylori* growth [22, 25, 26].

Based on generally available international data on antibiotic resistance in *H. pylori*, the level of sensitivity seems to vary internationally according to the general use of antibiotics and the antibiotic combinations applied in *H. pylori* therapy [26]. In addition, changes in the antibiotic sensitivity pattern of *H. pylori* may also serve as a sign of antibiotic interference with the human microflora. Use of these drugs should be carefully monitored. Increasing resistance should advocate reduced consumption and to target the use of antibiotics carefully to secure a successful outcome while preserving these valuable drugs for serious infections to come. Thus, continuous surveillance of antibiotic resistance in this *H. pylori* is warranted in order to secure optimal treatment options for gastric diseases patients and infectious disease patients in general. Finally, the data from the European surveillance study [22–27] indicate an increase in levofloxacin resistance is to be expected and the drug should be reserved primarily for treatment failures. A susceptibility check of the offending strain should definitely be performed to ensure *H. pylori* susceptibility to the drugs to be used for eradicating the bacteria even though such sensitivity testing is, for various reasons, done to a lesser extent than it should be [28].

The strength of this study is we have studied antibiotic resistance in *H. pylori* comparing a previous treatment-resistant group to a group newly diagnosed for *H. pylori* infection. Moreover, the potential of resistance to levofloxacin–the apparent drug of choice in treatment-resistant cases- has been studied. The weakness of the study is the lack of a continuous surveillance of antibiotic resistance in this microbe as indicated above. Therefore, a future ongoing study to secure optimal treatment options is highly warranted and especially focuses on levofloxacin resistance.

In conclusion, clarithromycin resistance was more frequently observed in patients resistant to *H. pylori* treatment than in patients newly diagnosed for *H. pylori* infection. None of the samples showed resistance to amoxicillin and levofloxacin. This implies that levofloxacin should be the drug of choice in cases resistant to ordinary triple regimens. When antibiotic therapy is planned, sensitivity testing of the offending strain should be performed to define the antibiotics to be used and thus increase the chance for successful eradication of the *H. pylori* infection.

## Supporting information

**S1 Protocol.**
(DOCX)

**S2 Protocol.**
(DOCX)

**S1 Table.**
(DOCX)

**S1 File.**
(SAV)

## Acknowledgments

We thank Odd Sverre Moen at Research Group Gastroenterology and Nutrition, UiT The Arctic University of Norway, Tromsø, Norway for expert technical assistance.

## Author Contributions

**Conceptualization:** Oddmund Nestegard, Eyvind Paulssen.

**Data curation:** Oddmund Nestegard, Kay-Martin Johnsen, Rasmus Goll.

**Formal analysis:** Oddmund Nestegard, Behrouz Moayeri, Sveinung Wergeland Sørbye, Kay-Martin Johnsen, Rasmus Goll, Jon Ragnar Florholmen, Kjetil K. Melby.

**Funding acquisition:** Oddmund Nestegard.

**Investigation:** Fred-Arne Halvorsen, Tor Tønnesen, Sveinung Wergeland Sørbye.

**Methodology:** Eyvind Paulssen, Rasmus Goll.

**Project administration:** Eyvind Paulssen.

**Supervision:** Jon Ragnar Florholmen, Kjetil K. Melby.

**Validation:** Sveinung Wergeland Sørbye, Jon Ragnar Florholmen, Kjetil K. Melby.

**Writing – original draft:** Oddmund Nestegard.

**Writing – review & editing:** Oddmund Nestegard.

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
