## [Decision Letter · Decision Letter 0]

1 Dec 2021

PONE-D-21-21216Helicobacter pylori resistance to antibiotics in patients 15 years after unsuccessful eradicationPLOS ONE

Dear Dr. Nestegard,

Thank you for submitting your manuscript to PLOS ONE. After careful consideration, we feel that it has merit but does not fully meet PLOS ONE’s publication criteria as it currently stands. Therefore, we invite you to submit a revised version of the manuscript that addresses the points raised during the review process.

We look forward to receiving your revised manuscript.

Kind regards,

Grzegorz Woźniakowski, Full professor, PhD, ScD

Academic Editor

PLOS ONE

Journal Requirements:

“NO”

Reviewers' comments:

Reviewer's Responses to Questions

**Comments to the Author**

1. Is the manuscript technically sound, and do the data support the conclusions?

Reviewer #1: Yes

Reviewer #2: Yes

2. Has the statistical analysis been performed appropriately and rigorously? 

Reviewer #1: Yes

Reviewer #2: Yes

3. Have the authors made all data underlying the findings in their manuscript fully available?

Reviewer #1: Yes

Reviewer #2: Yes

4. Is the manuscript presented in an intelligible fashion and written in standard English?

Reviewer #1: Yes

Reviewer #2: Yes

5. Review Comments to the Author

Reviewer #1: A research study was conducted on two groups of patients to describe the antimicrobial resistance pattern of H. pylori. Bacterial strains recovered from patients after unsuccessful treatment for H. pylori infection showed clarithromycin resistance more frequently than strains from patients with newly diagnosed H. pylori infection.

Minor revisions:

1- Include a brief statistical analysis section which summarizes the type of descriptive data presented. This section should also cite the statistical software used for the analysis.

2- Within the text, provide percentages corresponding to frequencies.

Reviewer #2: The article raises a public health problem, it is about the resistance to antibiotics in bacteria which, if still untreated, will be responsible at long term to gastric cancer.

The study made by this author is very interesting, because it is a monitoring of the resistance of the main antibiotics given in therapy for Helicobacter pylori eradication.

I congratulate the author on your work. However, I added in attach some comments and propositions to improve your article together.

Good continuation.

6. PLOS authors have the option to publish the peer review history of their article (what does this mean?). If published, this will include your full peer review and any attached files.

Reviewer #1: No

Reviewer #2: No

---

## [Author Response · Author response to Decision Letter 0]

12 Jan 2022

We have done changes in the article due to proposals from editor end reviewers.

That is the title of the article, abstract and keywords.

In introduction we have taken in relationship between H. pylori and gastric cancer. Also WHOs concerning about H. pylori and antibiotic resistance. There is a new reference for this.

In material and methods we have done changes as indicated from the reviewers and there is information about statistic there.

In results we have specified the inclusion and exclusion criteria.

In discussion we have added percentage of resistens for tetracycline.

In tables we have changed text and have put in sensitivity in percentage for each antibiotics.

MICs are in a Supporting Information file

---

## [Decision Letter · Decision Letter 1]

1 Mar 2022

Helicobacter pylori resistance to antibiotics before and after treatment: incidence of eradication failure.

ClinicalTrials.gov Identifier: NCT05019586

PONE-D-21-21216R1

Dear Dr. Nestegard,

We’re pleased to inform you that your manuscript has been judged scientifically suitable for publication and will be formally accepted for publication once it meets all outstanding technical requirements.

Kind regards,

Grzegorz Woźniakowski, Full professor, PhD, ScD

Academic Editor

PLOS ONE

Additional Editor Comments (optional):

Reviewers' comments:

Reviewer's Responses to Questions

**Comments to the Author**

1. If the authors have adequately addressed your comments raised in a previous round of review and you feel that this manuscript is now acceptable for publication, you may indicate that here to bypass the “Comments to the Author” section, enter your conflict of interest statement in the “Confidential to Editor” section, and submit your "Accept" recommendation.

Reviewer #1: All comments have been addressed

Reviewer #2: All comments have been addressed

2. Is the manuscript technically sound, and do the data support the conclusions?

Reviewer #1: (No Response)

Reviewer #2: Yes

3. Has the statistical analysis been performed appropriately and rigorously? 

Reviewer #1: (No Response)

Reviewer #2: Yes

4. Have the authors made all data underlying the findings in their manuscript fully available?

Reviewer #1: (No Response)

Reviewer #2: Yes

5. Is the manuscript presented in an intelligible fashion and written in standard English?

Reviewer #1: (No Response)

Reviewer #2: Yes

6. Review Comments to the Author

Reviewer #1: (No Response)

Reviewer #2: Some revisions are needed to the article be ready to be published. I add in attached my comments.

Good continuation.

7. PLOS authors have the option to publish the peer review history of their article (what does this mean?). If published, this will include your full peer review and any attached files.

Reviewer #1: No

Reviewer #2: No

---

## [Editor Report · Acceptance letter]

11 Apr 2022

PONE-D-21-21216R1 

Helicobacter pylori resistance to antibiotics before and after treatment: incidence of eradication failure. 

Dear Dr. Nestegard:

I'm pleased to inform you that your manuscript has been deemed suitable for publication in PLOS ONE. Congratulations! Your manuscript is now with our production department. 

Kind regards, 

on behalf of

Prof. Grzegorz Woźniakowski 

Academic Editor

PLOS ONE